# Bile Acids Activate NLRP3 Inflammasome, Promoting Murine Liver Inflammation or Fibrosis in a Cell Type-Specific Manner

**DOI:** 10.3390/cells10102618

**Published:** 2021-10-01

**Authors:** Theresa Maria Holtmann, Maria Eugenia Inzaugarat, Jana Knorr, Lukas Geisler, Marten Schulz, Veerle Bieghs, Mick Frissen, Ariel E. Feldstein, Frank Tacke, Christian Trautwein, Alexander Wree

**Affiliations:** 1Department of Hepatology and Gastroenterology, Charité University Medicine Berlin, Augustenburger Platz 1, 13353 Berlin, Germany; theresa.holtmann@charite.de (T.M.H.); jana.knorr@charite.de (J.K.); lukas.geisler@charite.de (L.G.); marten.schulz@charite.de (M.S.); frank.tacke@charite.de (F.T.); 2Department of Internal Medicine III, RWTH University Hospital Aachen, Pauwelsstraße 30, 52074 Aachen, Germany; m.euge.inzaug@gmail.com (M.E.I.); veerlebieghs@hotmail.com (V.B.); mfrissen@ukaachen.de (M.F.); ctrautwein@ukaachen.de (C.T.); 3Department of Pediatrics, University of California, San Diego, CA 92093, USA; afeldstein@ucsd.edu

**Keywords:** sterile inflammation, hepatic stellate cells, Kupffer cells, cholestasis, inflammagens, cholic acid, deoxycholic acid, lithocholic acid

## Abstract

Bile acids (BA) as important signaling molecules are considered crucial in development of cholestatic liver injury, but there is limited understanding on the involved cell types and signaling pathways. The aim of this study was to evaluate the inflammatory and fibrotic potential of key BA and the role of distinct liver cell subsets focusing on the NLRP3 inflammasome. C57BL/6 wild-type (WT) and *Nlrp3*^−/−^ mice were fed with a diet supplemented with cholic (CA), deoxycholic (DCA) or lithocholic acid (LCA) for 7 days. Additionally, primary hepatocytes, Kupffer cells (KC) and hepatic stellate cells (HSC) from WT and *Nlrp3*^−/−^ mice were stimulated with aforementioned BA ex vivo. LCA feeding led to strong liver damage and activation of NLRP3 inflammasome. Ex vivo KC were the most affected cells by LCA, resulting in a pro-inflammatory phenotype. Liver damage and primary KC activation was both ameliorated in *Nlrp3*-deficient mice or cells. DCA feeding induced fibrotic alterations. Primary HSC upregulated the NLRP3 inflammasome and early fibrotic markers when stimulated with DCA, but not LCA. Pro-fibrogenic signals in liver and primary HSC were attenuated in *Nlrp3*^−/−^ mice or cells. The data shows that distinct BA induce NLRP3 inflammasome activation in HSC or KC, promoting fibrosis or inflammation.

## 1. Introduction

Cholestasis is defined as an impairment in-bile flow and is present in many liver diseases. Causes are diverse and reach from congenital defects, genetic and autoimmune diseases, pregnancy, ischemia and toxic injuries to mechanical obstruction by gall stones or tumor compression [1,2]. Common histopathological features include hepatocellular injury, fibrosis, and sterile inflammation leading to cirrhosis in chronic stages. Unfortunately, limited therapeutic options are available, and a liver transplant is often indicated.

BA, the major organic solutes of bile, are considered crucial for the pathogenesis of cholestatic liver diseases [3,4]. BA are synthesized from cholesterol by hepatocytes and are secreted into the bile ducts, thereby reaching the small intestine and are partially reabsorbed in the terminal ileum [5]. Several studies have characterized cellular and molecular effects of retention of bile acids, including highly toxic (DCA, LCA) as well as non-toxic (CA, ursodeoxycholic acid (UDCA)) bile acids [6,7]. Most studies have focused only on direct cytotoxic effects in hepatocytes [8,9], but new evidence arises that pathologically elevated concentrations of bile acids during cholestasis are not high enough to directly cause any hepatocellular cell death on its own and that other cell types must be involved [3,10]. Focus shifted to bile acids as important signaling molecules and inflammagens [4,5,10,11], which can cause a sterile inflammation resulting in initial injury, infiltration of innate immune cells, and transdifferentiation of quiescent to myofibroblastic HSC [4,6]. Still, the initiators of the inflammatory and fibrogenic responses are poorly known, and insights can lead to new therapeutic strategies. 

The NLR (nucleotide-binding oligomerization domain (NOD) leucine-rich repeat containing receptors) family pyrin domain-containing 3 (NLRP3) inflammasome is the best understood member of the inflammasome family and plays an important role in many liver diseases with chronic, sterile inflammation [12,13]. A wide range of stimuli can activate the NLRP3 inflammasome, e.g., damage associated molecular patterns (DAMPS), pathogen associated molecular patterns (PAMPS), reactive oxygen stress (ROS), and Cathepsin B [14,15,16]. The activation is a two-step process that is primed by upregulation of pro-IL-1β, pro-IL-18 and pro-Caspase-1. In a second step, assembly of NLRP3 with pro-Caspase-1 leads to cleavage of the latter into its active form, which subsequently cleaves pro-IL-1β and pro-IL-18 into their mature cytokines [13,17,18].

There is evidence that the NLRP3 inflammasome and its downstream effectors are upregulated in cholestatic diseases in humans, e.g., biliary atresia, primary biliary cholangitis (PBC) and primary sclerosing cholangitis (PSC) [19,20,21]. Many murine studies showed an activation of the inflammasome due to exposure to different toxic bile acids [6,15,22,23]. Evidence arises that inhibition of NLRP3 by MCC-950 in mice lowers cholestatic injury and fibrosis in vivo [24], which may indicate therapeutic options.

Bile acid dependent mechanisms are still not fully understood as current studies are limited to experiments in hepatic cell lines, primary hepatocytes, cholangiocytes, peritoneal macrophages and bone marrow-derived macrophages (BMDMs) [15,23,25,26]; although, the NLRP3 inflammasome was shown to be functional in KC and HSC as well [27].

We aimed to test our hypothesis that in cholestasis bile acids act as cell-specific inflammagens by activation of the NLRP3 inflammasome in HSC and KC.

## 2. Materials and Methods

### 2.1. Animal Models

In this study, C57BL/6 wild-type (WT) and *Nlrp3*-deficient *(Nlrp3*^−/−^*)* (Jackson Laboratory, Bar Harbor, ME, USA, B6.129S6-Nlrp3^tm1Bhk^/J) mice were used. Both strains were cohoused in temperature and humidity-controlled environment with a 12 h light and dark cycle in accordance with the principles of the Guide for the Care and Use of Laboratory Animals. All procedures were approved by German local governmental authorities (Landesamt für Natur, Umwelt und Verbraucherschutz Nordrhein-Westfalen, LANUV NRW).

#### 2.1.1. Bile Acid Feeding

Six 12-week-old male or female animals of each strain were fed with either CA, DCA or LCA (Sigma-Aldrich, St. Louis, MO, USA) at a concentration of 0.5% in powder chow for seven days. Thereafter, mice were sacrificed, and liver tissue and serum were collected.

#### 2.1.2. Cell Isolation and Stimulation

For isolation of KC, HSC, and hepatocytes 20-week-old female or male mice were used. Briefly, mice were anesthetized by ketamine/xylazine injection and perfused in situ through the inferior vena cava with sequential Pronase E (0.4 mg/mL) and Collagenase D (0.8 mg/mL) (Sigma-Aldrich, St. Louis, MO, USA) solutions. Liver was removed and digested in vitro with Collagenase D (0.5 mg/mL), Pronase E (0.5 mg/mL) and DNAse I (0.02 mg/mL) (Sigma-Aldrich, St. Louis, MO, USA). After 20 min, tissue was filtered through a 70 µm mesh and hepatocytes were separated by centrifugation. Hepatocytes were washed and seeded into collagen-coated plastic tissue culture dish. The remainder of the cells were separated using a Nycodenz (PROGEN Biotechnik, Heidelberg, Germany) gradient centrifugation. HSC or KC were then washed and seeded separately onto plastic tissue dishes in DMEM or RPMI containing fetal serum and incubated at 37 °C with CO_2_ overnight. Next morning, medium was changed, and cells were washed and stimulated with CA, DCA, or LCA (100 µM) for 3 h followed by either Nigericin (10 µM) or ATP (5 mM) for 1 h.

### 2.2. Histology, Immunostaining and Sirius Red Staining

Liver tissues were fixed, embedded in paraffin, and processed on slides for hematoxylin-eosin (H&E) or Sirius Red staining. Primary monoclonal antibodies used to perform immunostaining were: MPO (ThermoFisher, Waltham, MA, USA), αSMA, anti-Collagen I, anti-IL-1β (Abcam, Cambridge, UK), anti-NLRP3 (AdipoGen, San Diego, CA, USA). The negative controls in all procedures omitted primary antibody. Cells were fixed with pre-chilled methanol and blocked with 5% BSA in PBS-T while the slides were deparaffinized and rehydrated in ethanol and the antigens were retrieved in citrate buffer pH 6.0 for 30 min at 95 °C or treated with 2% BSA 1´ Triton in TBS-T for 30 min at room temperature. Following overnight incubation with primary antibodies, HRP Alexa-Fluor 488 or Alexa-Fluor 546 conjugated secondary antibodies (ThermoFisher, Waltham, MA, USA) were applied. For color reaction of HRP, Streptavidin-peroxidase complex 3,3-diaminobenzidine tetrahydroxychloride was used as chromogen and the slides were counterstained with hematoxylin. Mounting solution containing DAPI (Vector Laboratories, Burlingame, CA, USA) was used to counterstain the nuclei for immunofluorescence. To measure Caspase-1 activity in liver tissue, cryosections were fixed with acetone, blocked with 5% BSA in PBS-T and then incubated with FLICA FAM-YVAD-FMK (FAM-FLICA^®^ Caspase-1 Assay Kit, ImmunoChemistry Technologies, Bloomington, MN, USA) for 2 h. The slides were then washed and DAPI containing mounting medium was applied. Pictures were taken with Axioimager Z1 using Axio Vision 4.2 software (Carl Zeiss, Jena, Germany) and analyzed using ImageJ software (U.S. National Institutes of Health, Bethesda, MD, USA). Confocal imaging was performed with LSM 710 confocal laser scanning microscope (Carl Zeiss) (Immunohistochemistry & Confocal Microscopy Facility, RWTH Uniklinik Aachen).

### 2.3. Real-Time PCR

Total RNA was isolated using PeqGold TriFast following manufacturer’s instruction (PeqLab, Erlangen, Germany). The reverse transcript (cDNA) was synthesized from total RNA using the iScript cDNA Synthesis kit (BioRad, Hercules, CA, USA). Real-time PCR quantification was performed using Fast Sybr-Green and QuantStudio 6 (Applied Biosystems, Waltham, MA, USA).

### 2.4. Flow Cytometry Analysis

After stimulation, HSC were harvested, permeabilized and incubated with Vimentin antibody conjugated to PE (R&D, Minneapolis, MN, USA), while KC were directly stained with F4/80 (Biolegend, San Diego, CA, USA), and Galectin3 (eBioscience, Waltham, MA, USA) antibody directly conjugated to FITC and PE, respectively. To assess Caspase-1 activity, FLICA FAM-YVAD-FMK (FAM-FLICA^®^ Caspase-1 Assay Kit, ImmunoChemistry Technologie) was used following manufacturer’s instructions. In order to measure reactive oxygen species production, cells were incubated with 2′,7′-dichlorofluorescein diacetate (DCFH-DA) (Sigma-Aldrich) for 30 min at 37 °C and analyzed by flow cytometry at 500 nm (BD FACS Canto II, Becton, Dickinson and Company, Franklin Lakes, NJ, USA).

### 2.5. ELISA

Quantification of IL-1β levels in supernatant was performed according to the manufacturer’s instruction (Murine IL-1β Standard ABTS ELISA Development Kit, Peprotech, Hamburg, Germany).

### 2.6. Fluorescence in Situ Hybridization (FISH)

After deparaffinization and rehydration, the specimens were permeabilized with PBS-0.3% Triton for 30 min and incubated in hybridization buffer at 65 °C for 20 min. Then, samples were heated up to 75 °C for 2 min and incubated with FITC-conjugated *Nlrp3* probe (Eurofins, Luxembourg City, Luxembourg; sequence: AGA TAC CAT ACG GTC CTC CTG ) overnight at 65 °C, washed with hybridization buffer and mounted with DAPI containing mounting medium.

### 2.7. Statistics

Analyses were performed with Graph Pad (version 7.0; Graph Pad, Graph Pad Software Inc., La Jolla, CA, USA). The significance level was set at α = 5% for all comparisons. Normal distribution was tested by D’Agostino-Pearson normality test. When data had approximately normal distributions, it was analyzed by using ANOVA. If data deviated from normal distribution, non-parametric Mann–Whitney test was used for two-group comparison. For experiments involving three or more groups, data were evaluated using Kruskal–Wallis test and Dunn’s Multiple Comparison Test. Unless otherwise stated, data are expressed as mean ± SEM.

## 3. Results

### 3.1. DCA and LCA Feeding Causes Liver Damage and Activation of NLRP3 Inflammasome

First, histopathologic damage in CA, DCA and LCA acid fed mice was assessed via hematoxylin and eosin staining. CA feeding did not have any deleterious effects in the liver and thus was considered as a negative control in the following (Figure 1A, Appendix A). In contrast, DCA feeding resulted in an intermediate level of damage with discrete infarcts and necrotic areas (Figure 1A). LCA-fed mice developed a severe phenotype with multiple infarcts and necrotic areas (Figure 1A). Consistently, serum transaminase levels (ALT, AST), levels of Alkaline phosphatase (ALP) as a cholestatic marker and Glutamate dehydrogenase (GLDH) for mitochondrial damage were within normal range for CA but mildly increased in DCA and severely increased in LCA (Figure 1B). No differences were observed between male or female mice. Liver–body weight ratio was elevated after LCA feeding compared to CA and DCA (Appendix A).

To determine the role of immune response in the development of cholestatic liver damage, NLRP3 inflammasome activation was evaluated. On mRNA levels of whole liver lysates, upregulation of *Il-1b* and *Nlrp3* in LCA fed mice was evident (Figure 1C). In both, DCA and LCA fed mice, an elevation of *Nlrp3* RNA positive cells in liver tissue stained by FISH compared to CA was found (Figure 1D). Moreover, staining with FAM-FLICA Caspase-1 activity assay verified an overall increase of active Caspase-1 in DCA fed mice. LCA feeding revealed a stronger activation and additional Caspase-1 positive cells at the margin of necrotic areas (Figure 1E).

### 3.2. Bile Acid induced Liver Damage Was Ameliorated in Nlrp3^−/−^ Mice

To verify, that the proven activation of NLRP3 inflammasome was not only a consequence but crucial for the observed liver damage by DCA and LCA, the same diets were fed to *Nlrp3*^−/−^ mice. As anticipated, CA feeding did neither harm in WT nor *Nlrp3*^−/−^ mice (Appendix A).

The intermediate phenotype of DCA was ameliorated in knockouts with reconstitution to normal liver morphology (Figure 2A) and almost normal serum transaminases (Figure 2E). Knockout mice showed decreased *Il-1b* mRNA levels and Caspase-1 activity compared to WT (Figure 2B,F). In addition, *Nlrp3*^−/−^ mice were protected against DCA induced fibrotic alterations as demonstrated by reduced SiriusRed stained areas (Figure 2D) and a decrease in mRNA expression levels of *alpha Smooth muscle actin* (α*Sma*), *Ctgf* and *Col1A1* as early markers of fibrosis (Figure 2H). However, no effect on immune cell invasion was observed as suggested by Myeloperoxidase (MPO) staining to assess for pro-inflammatory neutrophils and mRNA levels of *F4/80*, *F4/80, Cd68, Mcp1, Tnf, iNOS and IL-18* (Figure 2C,G). In conclusion, despite moderate liver injury and lack of immune cell invasion, DCA feeding promoted a strong fibrotic response that was ameliorated in *Nlrp3*^−/−^ mice.

Similarly, *Nlrp3*^−/−^ mice showed attenuated phenotype after LCA feeding in comparison to WT. Liver tissue revealed fewer necrotic areas and bile infarcts, albeit a normal morphology was not observed (Figure 3A). Consistently, serum transaminase levels trend to decrease but were not within normal range (Figure 3E). Analyzing NLRP3 components in the knockouts, we found a significant decrease in *Il-1b* mRNA levels (Figure 3F) and diminished Caspase-1 activity in overall liver tissue and at the margin of necrotic areas (Figure 3B). Small fibrotic alterations on histopathological and mRNA levels tend to remain, regardless of NLRP3 deficiency (Figure 3D,H). In contrast, we found fewer invasion of MPO positive cells, suggesting a less pro-inflammatory environment after LCA feeding in knockout mice (Figure 3C). This observation was confirmed by analysis of mRNA expression: markers of cell infiltration and activation (*F4/80, Cd68, Mcp1*) and markers of a proinflammatory phenotype of macrophages (*Tnf, iNos, Il-18*) were significantly decreased (Figure 3G). 

### 3.3. DCA, but Not LCA, Activates HSC to a Fibrotic Phenotype

To investigate how DCA and LCA can both activate the NLRP3 inflammasome in whole liver lysate but lead to such distinct patterns of fibrotic or inflammatory liver damage, a cell-specific model was established. Primary hepatocytes, HSC and KC were stimulated separately ex vivo with different bile acids.

Neither CA, DCA nor LCA stimulation affected basal transaminase levels or NLRP3 inflammasome components in primary hepatocytes (Appendix A). 

HSC from WT were isolated and primed with CA, DCA or LCA as a first signal. In addition, adenosine triphosphate (ATP) was used as a common second signal of NLRP3 activation. As anticipated, CA did not cause any changes in NLRP3 inflammasome activation or fibrotic phenotype compared to non-stimulated HSC and neither did LCA (Appendix A). Although the revealed damage in LCA fed mice was severe, this effect could not be explained by direct activation of HSC. 

In contrast, primary HSC stimulated with DCA showed an activation of the NLRP3 inflammasome as indicated by upregulation of mRNA levels of *Nlrp3* and *Il-1b* (Figure 4A) and elevated IL-1β levels in the supernatant of the cells (Figure 4B). In addition, Caspase-1 activity was measured by flow cytometry as it is part of a functional NLRP3 inflammasome. DCA followed by ATP stimulation induced an increase in Caspase-1 activity in HSC when normalizing the mean fluorescence intensity to basal conditions (Figure 4D).

In the immunofluorescent staining, an increase of NLRP3 and the formation of IL-1β vesicles after stimulation with DCA and ATP was noticed (Figure 4C). Furthermore, a morphological change from quiescent HSC to more active, profibrotic, star-like shape HSC was observed. This effect was corroborated by increased deposition of Collagen I (Figure 4F) and upregulated mRNA expression of *Ctgf* and *Tgfb* when stimulated with DCA (Figure 4E). At the protein level, FACS analysis revealed a significant increase in expression of Vimentin—an early fibrotic marker of HSC (Figure 4G).

### 3.4. Activation of HSC by DCA Was Abolished in Nlrp3^−/−^

To assess whether the observed activation of NLRP3 inflammasome was the crucial step in evolving a fibrotic phenotype, primary HSC from *Nlrp3*^−/−^ were stimulated with DCA and ATP as well. After stimulation of the knockout cells, neither mRNA levels of *Il-1b* nor protein levels were elevated when stimulated with DCA (+ATP) (Figure 5A–C). In contrast to WT cells, activity of Caspase-1 remained at a basal level in *Nlrp3*^−/−^ HSC (Figure 5D). No change of shape was observed, and Collagen I expression was not upregulated in *Nlrp3*^−/−^ HSC (Figure 5E). Moreover, neither *Ctgf* and *Tgfb* mRNA levels (Figure 5E), nor Vimentin expression was increased after DCA (+ATP) stimulation (Figure 5G). These findings indicate that the observed activation of NLRP3 by DCA seems to be a crucial step in the development of a fibrotic HSC phenotype. Furthermore, the observation that only DCA activates the HSC, but not LCA, may explain why attenuation of fibrotic alterations was observed in knockouts fed with DCA but not LCA.

### 3.5. LCA Stimulates NLRP3 Activation in KC and Provokes a Pro-Inflammatory Phenotype

Next, KC from WT and *Nlrp3*^−/−^ mice were isolated to investigate cell specific effects after stimulation with BA. Nigericin was used as a common second signal for NLRP3 activation. No effect after stimulation with CA or DCA was observed, neither on the NLRP3 inflammasome nor on the phenotype of primary KC (Appendix A). 

In contrast, LCA induced activation of the NLRP3 inflammasome as shown by upregulated mRNA expression of *Nlrp3* and *Il-1b* after LCA stimulation (Figure 6B), and elevated protein levels of intra- and extracellular IL-1β after LCA (+ Nigericin) (Figure 6A,C). mRNA levels of *Il-18* only trend to increase, but this difference was not significant. In addition, not only the activity of Caspase-1 was enhanced after LCA + Nigericin stimulation, but also the percentage of Caspase-1/PI double positive cells (Figure 6D). This indicates an increase NLRP3 inflammasome driven pyroptotic cell death as a response to LCA and Nigericin treatment. 

To determine how LCA can lead to inflammasome activation, ROS production, a well-known activator of NLRP3 inflammasome was assessed. By comparing fluorescence of DCF in FACS, an increase of ROS production after stimulation with LCA alone and in addition to Nigericin was observed in comparison to basal levels (Figure 6E). This suggests that activation of NLRP3 inflammasome in KC after LCA + Nigericin can be mediated via generation of ROS. In addition, the changes of KC phenotype after LCA (+ Nigericin) treatment were characterized. An upregulation of pro-inflammatory mRNA markers as *Tnf, iNos, Ccr8, Ccl22* and *ActivinA* was quantified, whereas *Arg1* as a marker for anti-inflammatory macrophages was downregulated in LCA stimulated KC (Figure 6F). This observation was further validated on protein level by Galectin 3 as a proinflammatory macrophage marker. Galectin 3-positive cells in FACS were increased after LCA treatment followed by Nigericin when compared to the medium control (Figure 6G).

### 3.6. Activation of NLRP3 Inflammasome and Pro-Inflammatory Phenotype Was Blocked in Nlrp3^−/−^ KC Despite LCA Treatment

KC from *Nlrp3*^−/−^ mice were isolated and stimulated with LCA followed by Nigericin to check if NLRP3 inflammasome activation was crucial for development of the observed proinflammatory phenotype. 

In contrast to WT, the knockout KC failed to upregulate mRNA levels of *Il-1b* and *Il-18* (Figure 7B) after exposure to LCA (+Nigericin) and protein levels of IL-1β remained at a basal level (Figure 7A,C). Consistently, no increase of pyroptotic, Caspase-1/PI double positive cells was found after stimulation with LCA + Nigericin (Figure 7E). *Nlrp3*^−/−^ KC showed an increased ROS production after LCA treatment alone and with Nigericin, indicating that NLRP3 deficiency could not prevent from oxidative stress. 

However, *Nlrp3*^−/−^ KC did not respond with upregulation of proinflammatory markers to ROS as expression of *Tnf, iNos, Ccr8, Ccl22* and *ActivinA* remained at basal level after LCA treatment (Figure 7G). Moreover, protein levels of Galectin 3 were not increased after stimulation with LCA and Nigericin (Figure 7F). 

These findings suggest, that although LCA still triggers ROS generation in *Nlrp3*^−/−^ KC, its downstream harmful effector—the NLRP3 inflammasome—is crucial for the evolving proinflammatory phenotype and that its blockage can prevent inflammation generated by ROS.

## 4. Discussion

Our data uncovers the important role of bile acids as inflammagens during cholestasis, which activate NLRP3 inflammasome and thereby contribute to liver damage and early fibrosis in a cell type-specific manner.

Before, bile acids were thought to be directly cytotoxic for cells in the liver, especially for hepatocytes [8,9,28,29]. The main focus was on their detergent properties. From this point of view, bile acids were sorted from low to highly toxic acids with common sequence of ursodeoxycholic < cholic < chenodeoxycholic < deoxycholic < lithocholic acid [6,7,29]. Necrosis, and to a smaller part, apoptosis were considered to be crucial for the development of cholestatic liver injury once hepatocytes were brought in contact with high concentrations of bile acids [4].

New evidence has arisen that in vivo concentrations of accumulated bile acids in experimental models and patients with cholestasis are not high enough to cause necrosis or apoptosis via their direct cytotoxic effects to the previous thought extent [3,4,30,31]. Total serum levels of bile acids seem to peak at maximum 300 µM during extensive cholestasis in vivo in rats, mice and humans, when most studies considered lower levels [3,10,32,33], for example 5–15 µM [5]. Woolbright et al. showed that overall concentrations of distinct BA range from 1 to 22 µM (for glycochenodeoxycholic acid, GCDC) in serum. However, they described cytotoxicity in human hepatocytes not until GCDC is used at a concentration of 1000 µM [34]. Galle et al. revealed cytotoxicity only when hepatocytes were exposed to ≥ 500 µM of GCDC [28]. Billington et al. observed lysis by taurocholic acid (TCA) at 12–16 mM and glycodeoxycholate at 1.5–2 mM [35]. Having a closer look, the used concentrations were far off the margin for physiologically or pathologically measured concentrations of bile acids as they exceed the 2- to 70-fold of observed serum levels.

Our study contradicts the claims of older studies that propose a direct cytotoxic effect especially on hepatocytes as a major injury mechanism. CA, DA and LCA at levels that recapitulate in vivo concentrations (100 µM) lack cytotoxicity in hepatocytes (Appendix A). This finding is corroborated by Penman et al. and Allen et al., who described no cytotoxic effect on HepaRG cells or primary hepatocytes when exposed to BA in concentrations typically found in serum [10,36]. 

Hepatocytes have a functional NLRP3 inflammasome, are cruel for bile acid metabolism, and react to cytokines released from surrounding cells. Although hepatocytes did not activate the NLRP3 inflammasome after in vitro stimulation with BA, they may still play a key role in later stages of cholestatic liver injury. To address NLRP3 activation in hepatocytes when interacting with activated KC or HSC, co-culturing experiments will be needed.

Still, there is evidence that certain bile acids are harmful in pathological concentrations and can cause liver damage during cholestasis.

Recent studies emphasized BA as important signaling molecules that lead to liver damage not by cytotoxicity but by provoking an inflammatory environment [5]. Bile acids have been postulated to be inflammagens [30] that can cause sterile inflammation, leading to initial injury, infiltration of innate immune cells, and transdifferentiation of quiescent HSC to myofibroblasts—and in the long term, to liver fibrosis.

Those more diverse BA-activated signaling pathways and involved mediators of inflammation are still under investigation. Certain groups focused on the role of invasion of neutrophils in cholestasis [4,10,37,38], others focused on formation of ROS [39,40,41,42]. Besides, NLRP3 inflammasome became an object of interest, as its activity is upregulated in cholestatic liver injury in PSC, PBC, and biliary atresia [19,20,21].

In this study, we showed how DCA and LCA act as inflammagens by activating the NLRP3 inflammasome when fed to mice and when primary HSC or KC are exposed to them. As a result of LCA stimulation, a sterile inflammation was initiated by NLRP3 inflammasome, where augmented infiltration of immune cells as well as upregulation of pro-inflammatory markers were observed. DCA activated the NLRP3 inflammasome, leading to early fibrosis in livers. Deficiency of NLRP3 ameliorated the damage, indicating that NLRP3 inflammasome activation can be considered as an important inflammagen pathway triggered by BA. 

The next conclusion of our study was that bile acids have distinct effects on specific cell types in the liver. 

In the past, the main focus was on how bile acids act as inflammagens in hepatocytes, cholangiocytes or inflammatory cells [38]. Consensus was built, that isolated hepatocytes upregulate proinflammatory signaling if induced by different bile acids, especially by TCA or TCA, ßMCA and TßMCA collectively; expression and secretion of cytokines such as chemokine (C-X-C motif) ligand 1 (CXCL1) and CXCL2 as well as expression of intercellular adhesion molecule 1 (ICAM-1) were found to be increased [10,30].

Little was known about the role of KC and HSC, although these two liver cell types are crucial in the development of cholestatic liver injury and fibrosis [27,31]. 

Recently, our group showed that HSC express a functional NLRP3 inflammasome, which once activated, provokes morphological changes towards myofibroblasts with upregulation of fibrotic markers on protein and mRNA levels. Moreover, HSC specific gain-of-function mutation of NLRP3 in mice results in early onset of liver fibrosis [27]. 

This study suggests that the NLRP3 inflammasome of HSC can also be activated by DCA as a first signal followed by ATP. An increase of fibrotic markers on protein and mRNA levels was revealed and IL-1β was excreted into the supernatant. Early fibrotic stages were developed after feeding mice with DCA and could be reversed when NLRP3 was knocked out. In contrast, no change of fibrotic phenotype was observed between WT and *Nlrp3*^−/−^ when fed with LCA or CA in vivo and consistently the NLRP3 inflammasome in HSC was not activated when exposed to LCA or CA in vitro. Although the time period of BA feeding and in vitro stimulation is not long enough to cause severe fibrosis, the observed early fibrotic alterations can be the initiator for further changes up to liver cirrhosis with clinical impact.

KC as resident liver macrophages and part of the innate immune system are known to express a functional NLRP3 inflammasome. Thus, it was not surprising that the pronounced activation of NLRP3 inflammasome in whole liver lysate after LCA feeding was reproducible in primary KC stimulated with LCA. Once the inflammasome was activated, KC secreted IL-1β, expressed proinflammatory genes known for chemoattraction and showed an increase in pyroptosis. This activation of KC can cause a vicious cycle with formation of an inflammatory environment and attraction of more immune cells, as indicated by MPO-staining in LCA fed mice. KC did not react to that extent when stimulated with DCA, which may explain why invasion of immune cells was a less common feature in the livers from DCA-fed mice.

The exact pathways of NLRP3 activation by DCA and LCA in HSC and KC remain unclear. *Nrlp3*^−/−^ KC still showed increased ROS production after LCA stimulation (Figure 7); thus, ROS production may be a mediator between BA stimulation and NLRP3 activation. However, how BA trigger ROS production remains unanswered. 

BA receptors became focus of interest for drug targets in cholestatic diseases. Especially, Farnesoid X receptor (FXR) as a nuclear receptor, and Takeda G protein coupled receptor 5 (TGR5) as a membrane bound receptor are extensively investigated [43]. It is possible that bile acids trigger ROS production and inflammasome activation after binding to FXR or TGR5 [44]. Differences in the response of HSC, KC and hepatocytes to distinct bile acids may be due to different receptor profiles [45]. Moreover, HSC do not take up bile acids, which can lower the impact of intracellular BA receptors for HSC [46].

In summary, BA are a heterogenous group that should not be categorized by their detergent properties from low to highly toxic bile acids. Our study demonstrated that they have distinct abilities to trigger NLRP3 inflammasome in different cell types, thereby resulting in proinflammatory or profibrotic environments. LCA addresses KC leading to an inflammatory response, and DCA targets HSC resulting in their fibrotic activation. In addition, not only harmful, but also beneficial effects of bile acids have to be considered. UDCA and obeticholic acid have important therapeutic implications in cholestatic and fatty liver disease. BA are not only cytotoxic but play a complex role as mediators in liver damage and homeostasis, and their cell-specific effect should be examined more deeply.

In contrast to our results, several studies reported an inhibitory effect of BA on NLRP3 inflammasome activation. Guo et al. showed a decrease of IL-1β production and caspase maturation after LCA treatment in LPS-stimulated BMDM [47]. They stated that inhibition may be due to elevated ubiquitination of NLRP3 via the TGR5-cAMP-PKA axis. The disparity to our study could result from differences in used cell types and design of in vitro experiments. Guo et al. used BMDM, not primary liver cells, and stimulated additionally with LPS (±Nigericin) in every condition. Data from BA stimulation without LPS treatment are lacking and thereby comparisons to our results are difficult. Still, the appearance of conflicting results cannot be negated and a bidirectional role of NLRP3 inflammasome during cholestatic liver injury is conceivable. For example, Frissen et al. showed that *Nlrp3*^−/−^ mice develop increased liver injury in acute cholestasis but decreased liver injury and inflammation in chronic cholestasis [48].

BA are increasingly being considered as not just lipid solubilizers or cytotoxic detergents but as important signaling molecules; therefore, it is not surprising that recent focus was placed on BA-induced pathways as therapeutic targets in cholestatic liver disease [5]. 

Our experiments showed a protective effect of NLRP3 knockout in BA-induced liver damage and fibrosis in our in vivo and in vitro models. Lately, many pharmacological inhibitors of the NLRP3 inflammasome have been developed. Several of these drugs directly target the NLRP3 protein, whereas others block other components or downstream effectors [49]. MCC-950 is thus far the best investigated, direct, potent, selective and small-molecule inhibitor of NLRP3 and was shown to ameliorate NASH fibrosis [50] and liver damage in chronic stages of cholestasis in mice [24,48]. Although these results are promising, side effects of MCC-950 are not fully investigated and its pharmacokinetics characteristics do not apply for once-a-day oral dosing [50]. Further, second generation NLRP3 inhibitors are needed, and our cell-specific observations may pave the ground for a targeted blockage of NLRP3 in HSC and KC, such that physiological functions of NLRP3 in other liver and non-liver cells can be maintained.

## Figures and Tables

**Figure 1 cells-10-02618-f001:**
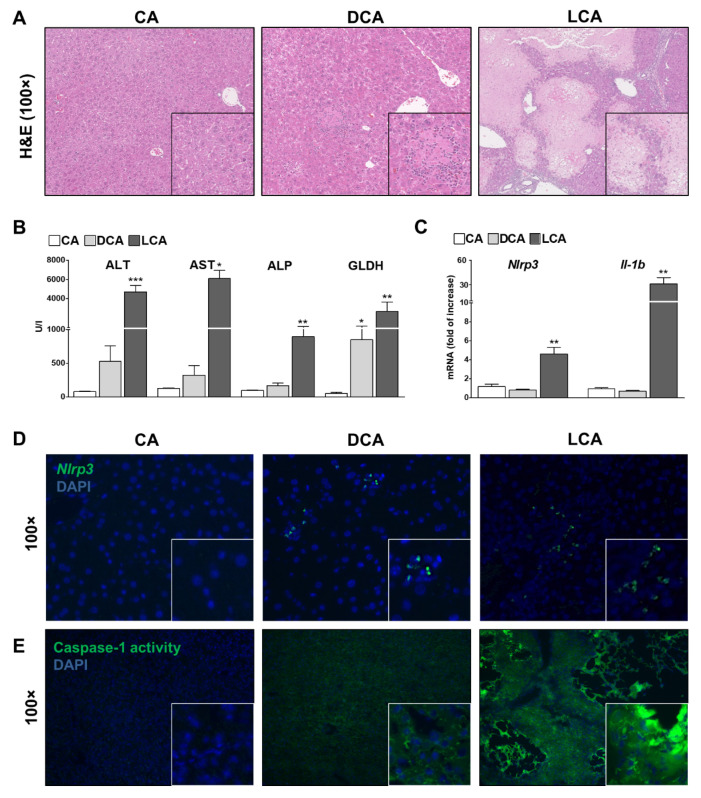
Feeding of DCA and LCA (0.5%) for seven days. H&E stainings reveal intermediate or severe liver damage in murine liver after feeding with DCA or LCA (**A**), confirmed by mildly or severely elevated levels of transaminases, ALP and GLDH (**B**). LCA feeding increased mRNA levels of *Nlrp3* and *Il-1b* (**C**). FISH shows a punctual upregulation of *Nlrp3* in DCA and LCA fed mice (**D**). Caspase-1 activity is mildly elevated after DCA feeding and strongly elevated after LCA feeding (**E**). CA feeding did not have any deleterious effects. * *p* < 0.05; ** *p* < 0.01; *** *p* < 0.001, n = 6.

**Figure 2 cells-10-02618-f002:**
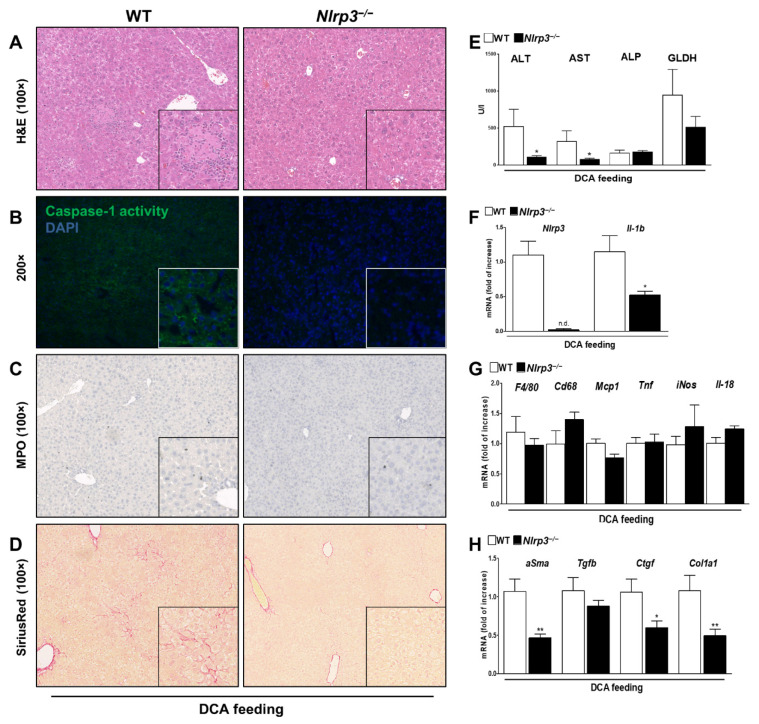
*Nlrp3**^−/−^* mice developed less liver damage and fibrosis after DCA feeding. Intermediate liver damage after DCA feeding was ameliorated in *Nlrp3*^−/−^ mice as indicated by H&E staining (**A**) and lower transaminase levels (**E**). *Nlrp3*^−/−^ mice show reduced Caspase-1 activity and *Il-1b* mRNA levels (**B**,**F**), resulting in attenuated fibrotic alterations as shown in SiriusRed stained liver sections (**D**) and decreased mRNA levels of fibrotic markers (*αSma, Ctgf, Col1a1*) (**H**). *Nlrp3*^−/−^ did not affect MPO positive cells and mRNA levels of *F4/80, Cd68, Mcp1, Tnf, iNOS and IL-18* as markers of immune cells invasion (**C**,**G**). * *p* < 0.05; ** *p* < 0.01; n.d.: not detected; n = 6.

**Figure 3 cells-10-02618-f003:**
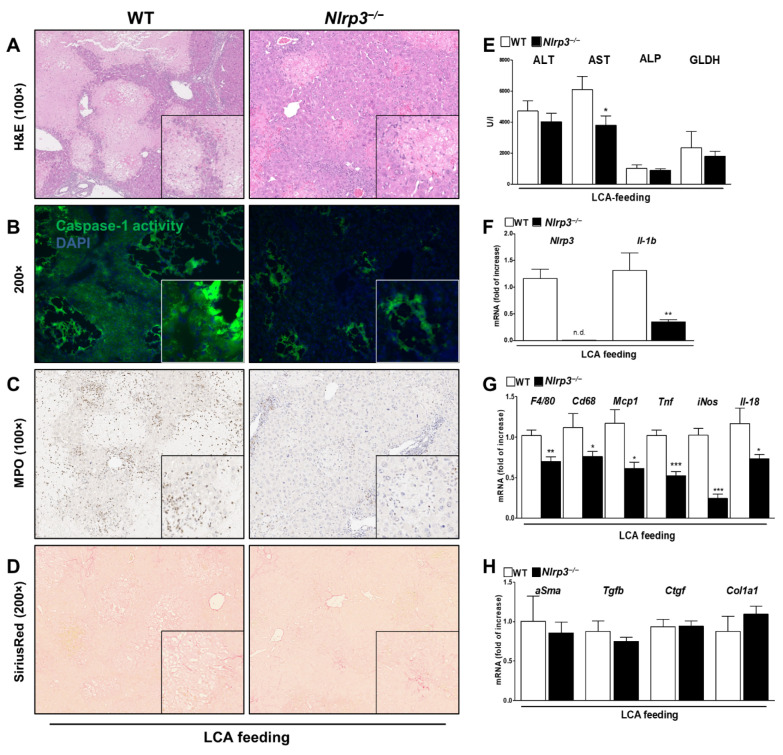
*Nlrp3*^−/−^ ameliorated severe liver damage and proinflammatory milieu after LCA feeding. *Nlrp3*^−/−^ attenuated severe liver damage after LCA feeding with smaller necrotic areas, fewer bile infarcts and lowered transaminase levels (**A**,**E**). However, normal morphology was not obtained. *Nlrp3*^−/−^ mice revealed lower Caspase-1 activity and *Il-1b* mRNA levels (**B**,**F**). Despite NLRP3 deficiency fibrotic markers remained the same (**D**,**H**), whereas infiltration of inflammatory, MPO-positive cells decreased (**C**). *Nlrp3*^−/−^ inhibited a proinflammatory environment after LCA feeding as indicated by decreased mRNA markers of immune cell invasion (*F4/80, Cd68, Mcp1*) and proinflammatory macrophages (*Tnf, iNos, Il-18*) (**G**). * *p* < 0.05; ** *p* < 0.01; *** *p* < 0.001; n.d.: not detected; n = 6.

**Figure 4 cells-10-02618-f004:**
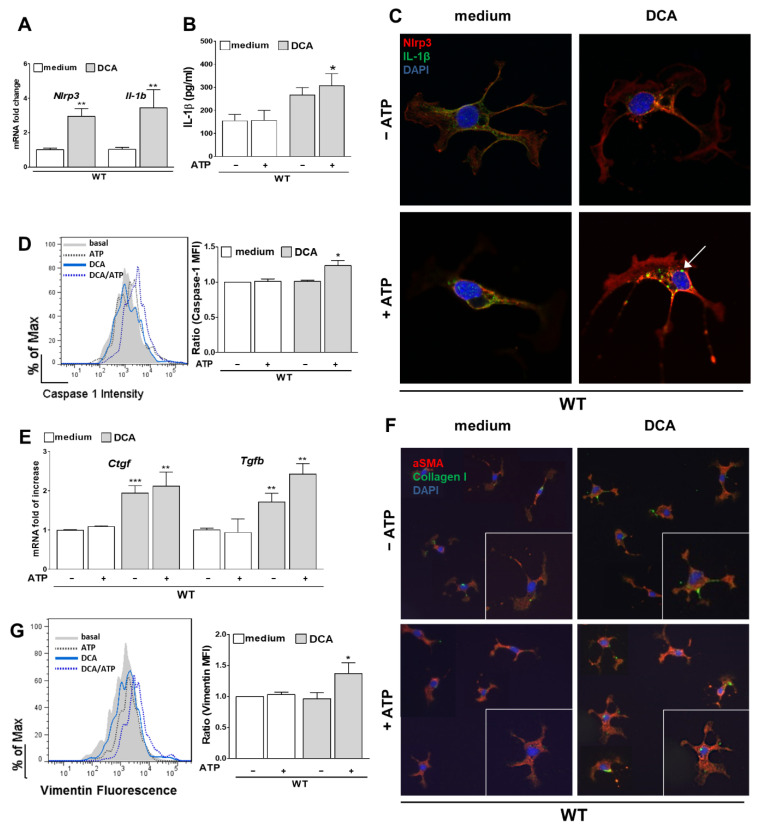
DCA activates the NLRP3 inflammasome in HSC leading to early fibrosis. Stimulation with DCA (+ ATP) activates the NLRP3 inflammasome in HSC as indicated by elevated mRNA levels of *Nlrp3* and *Il-1b* (**A**), increased levels of IL-1β in the supernatant (**B**) and activity of Caspase-1 (**D**). Microscopically, not only an upregulation of NLRP3 and formation of IL-1β vesicles were observed after immunofluorescent staining (**C**, arrow) but also a change of morphology to a more active, star-like shape with higher deposition of Collagen I (**F**). Upregulated mRNA levels of *Ctgf* and *Tgfb* (**E**) and an increase of the mean fluorescence intensity of Vimentin (**G**) as an early fibrotic protein marker confirmed this effect after DCA (+ATP) stimulation. * *p* < 0.05; ** *p* < 0.01; *** *p* < 0.001.

**Figure 5 cells-10-02618-f005:**
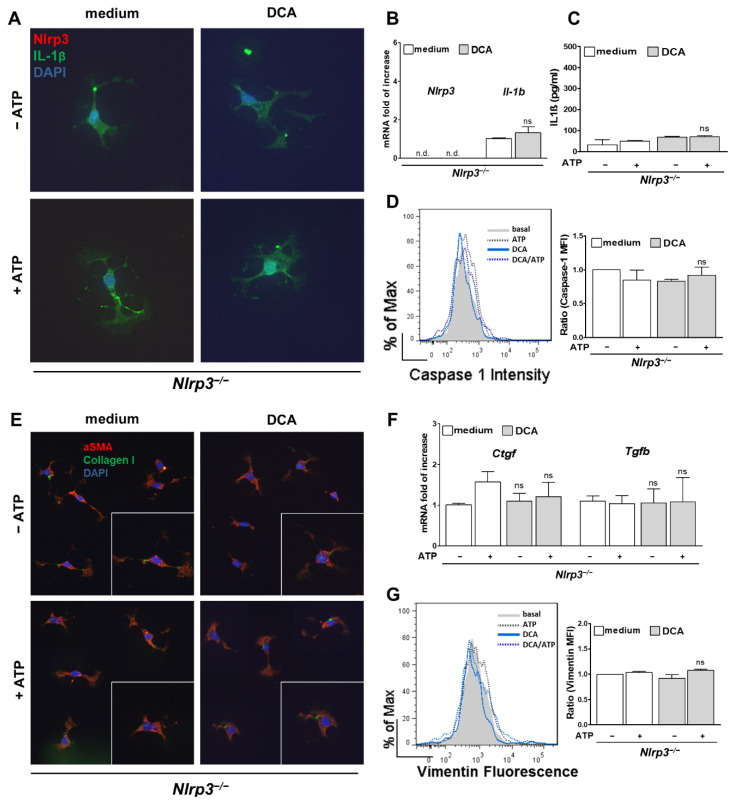
Stimulation of HSC from *Nlrp3*^−/−^ mice with DCA led to no fibrotic phenotype. HSC from *Nlrp3*^−/−^ mice showed no NLRP3 inflammasome activation after stimulation with DCA (+ ATP) as shown by stable mRNA levels of *Il-1b* (**B**), equal concentrations of IL-1β in the supernatant (**C**), omitted formation of IL-1β vesicles (**A**) and equivalent activity of Caspase-1 in flow cytometry analysis (**D**) when compared to basal levels. As a result, HSC were not activated to a fibrotic phenotype, as they failed to change their shape and deposit Collagen I (**E**) after DCA (+ ATP) stimulation. Early fibrotic markers such as Vimentin (**G**) and mRNA levels of *Ctgf* and *Tgfb* (**F**) remained the same. n.d.: not detected; ns: not significant.

**Figure 6 cells-10-02618-f006:**
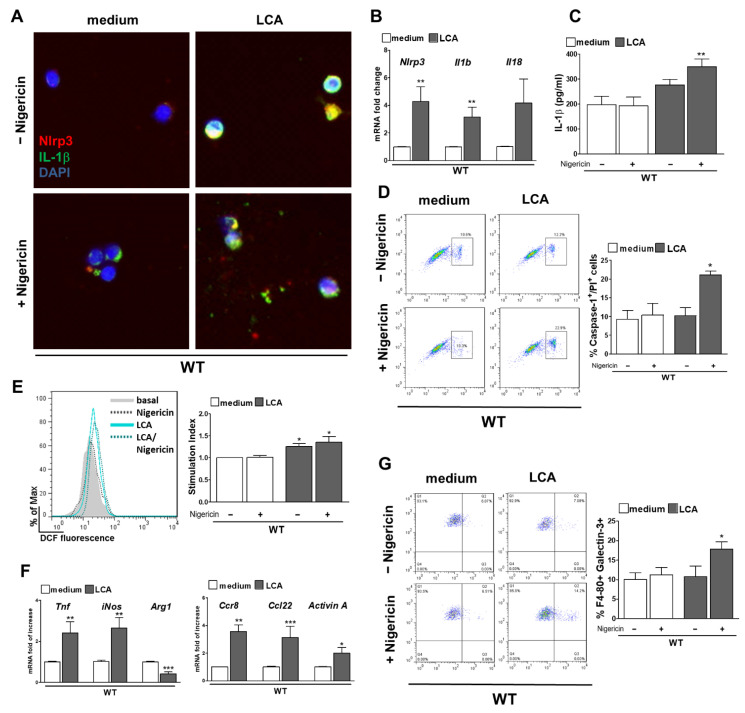
LCA stimulates NLRP3 activation in KC and provokes a pro-inflammatory phenotype. Stimulation of KC with LCA (+ Nigericin) activates the NLRP3 inflammasome as validated by increase of mRNA of *Nlrp3*, *Il-1b* and *Il-18* (**B**) and elevated levels of intra- and extracellular IL-1β (**A**,**C**). After exposure to LCA + Nigericin more Caspase-1/PI double positive cells were noted in FACS (**D**), which indicates pyroptotic cell death. Assessing possible pathways of NLRP3 activation, measurement of DCF fluorescence in KC showed an increase of ROS production after LCA (+ Nigericin) (**E**). Markers of proinflammatory macrophages (*Tnf, iNos, Ccr8, Ccl22* and *ActivinA*) increased after LCA stimulation, whereas expression of *ArginaseA* as marker for anti-inflammatory macrophages was decreased (**F**). More Galectin 3-positive KC were found after stimulation with LCA + Nigericin compared to medium (**G**). * *p* < 0.05; ** *p* < 0.01; *** *p* < 0.001.

**Figure 7 cells-10-02618-f007:**
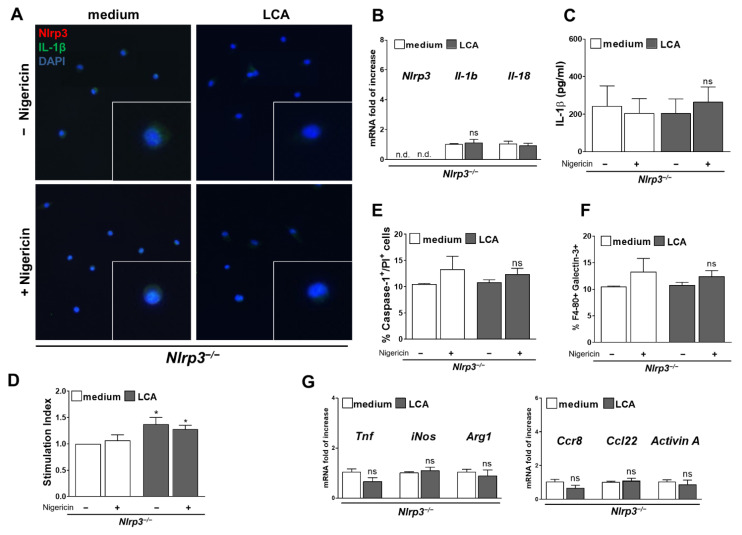
LCA failed to activate the NLRP3 inflammasome in *Nlrp3*^−/−^ KC provoking no proinflammatory phenotype. Stimulation of *Nlrp3*^−/−^ KC with LCA (+Nigericin) did not activate the NLRP3 inflammasome with no increase of mRNA expression of Il-1b (**B**) and no change in neither intra- nor extracellular IL-1β protein levels (**A**,**C**). Percentage of pyroptotic, Caspase-1/PI double positive cells was the same after exposure to LCA + Nigericin in *Nlrp3*^−/−^ KC compared to basal levels (**E**). Still, LCA provoked formation of ROS in *Nlrp3*^−/−^ KC as comparison of DCF fluorescence showed (**D**). However, KC from *Nlrp3*^−/−^ mice showed no increase of proinflammatory mRNA markers (Tnf, iNos, Ccr8, Ccl22, ActivinA) after LCA stimulation (**G**) and percentage of Galectin 3-positive cells remained at a basal level (**F**). * *p* < 0.05; n.d.: not detected; ns: not significant.

## Data Availability

Data is contained within the article or Appendix A.

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
