# Peer review of "Bile Acids Activate NLRP3 Inflammasome, Promoting Murine Liver Inflammation or Fibrosis in a Cell Type-Specific Manner"

_cells, 2021, doi:10.3390/cells10102618_

Round 1

Reviewer 1 Report

The work by Holtmann TM and coworkers studies the roles of bile acids in activating NLRP3 in mice and in primary hepatocytes, macrophages, and hepatic stellate cells (HSC). The authors found that LCA induces inflammation responses in WT mice and in primary macrophages, whereas DCA induces fibrotic response in WT mice and in HSC. These effects were all alleviated with NLRP3 deficiency. While the study addresses a potentially interesting role of bile acids in inflammation and fibrosis in types of hepatic cells, it has some methodological defects, including mixed gender, short term of treatments, and missing control groups.

Major concerns:

  1. Marked gender disparity in bile acid homeostasis as well as in cholestasis exists in both humans and rodents. However, the study used both male and female mice without further clarification. It is difficult to evaluate the data generated on a mixed gender background.
  2. Both in vivo and in vitro experiments were performed within a short period (7 days for mice and 3 hours for cells). The development of cholestatic liver injury, especially fibrosis, usually takes years in humans. It is suggested to justify the relevance of the study to the clinical observations.
  3. Figures 2 and 3. Please include CA groups in the figures instead of presenting in the suppl Fig.1. Without such controls, it is hard to estimate the changes. For example, the mRNA levels of Nlrp3 and fibrotic genes, as they were all presented as fold changes. Also, it would be better to include the data from mice fed on chow diet only.
  4. Hepatocytes are responsible for bile acid biosynthesis, secretion, and re-absorption. Although in the study, treatment of bile acid for 3 hours, did not alter Nlrp3 or Il-1b expression, it cannot exclude the involvement of hepatocytes in bile acid - induced inflammation and/or fibrosis. The authors need to discuss this in the manuscript. In fact, a co-culture study of hepatocytes with macrophages or HSC would provide more information reflecting what is happening in the liver upon bile acid stimulation.
  5. Some studies have reported that bile acids inhibit NLRP3 inflammasome (Immunity. 2016, 45: 802-816). The authors need to further discuss this issue.

Minor concerns:

  1. What about the livers weight after bile acid treatments?
  2. Will bile acid treatment alter de novo bile acid synthesis?
  3. Please number the figures in the order that they were mentioned in the manuscript. For example, suppl Fig. 4 was mentioned earlier than suppl Fig. 2 and 3.

Author Response

Thank you very much for your comments and input. Please see the attachment for our response.

Reviewer 2 Report

Holtmann et al. present a concise study that addresses an important topic regarding the function of bile acids as mediators of hepatic inflammation and fibrosis. The authors address what cell types found in the liver may account for inflammation and fibrosis associated with bile acid injury, and whether the NLRP3 inflammasome is required for this response. Using in vivo models and isolated hepatic stellate cells (HSC) and Kupffer cells (KC), the authors convincingly demonstrated that these cell types distinctly respond to the secondary bile acids lithocholic acid (LCA) and deoxycholic acid (DCA) via the infllammasome, with LCA promoting an inflammatory response in KC, while DCA induced the expression of fibrosis related genes in HSC. The authors also demonstrate, using nlrp3-/- mice, that the inflammasome is required for bile acid-induced responses of both KC and HSC. The authors further implicate reactive oxygen species (ROS) as a potential mediator of KC inflammasome activation. The manuscript is well-written, presented in a logical manner, and the figures support the authors conclusions. The manuscript could be improved by the following:

The authors do not address the mechanisms of bile acid-induced elevation of ROS or inflammasome activation in KC or HSC. Whether the bile acids are interacting with the respective cell types through unique bile acid receptors, such as TGR5 and FXR, to initiate the disparate responses was not addressed. While experimentally likely outside the scope of the current manuscript, the authors should discuss hepatic bile acid receptor expression and how bile acid receptors may be involved in the outcomes of their experiments.

The in vivo data would be strengthened if the authors showed both fibrosis and inflammatory gene expression (panel H) in both DCA (fig 2) and LCA (fig 3) feeding experiments.

It is unclear why the same second signals of NLRP3 activation were not used for both HSC and KC. The authors should either repeat the assays to include both second signals or justify use of one versus the other depending on cell type.

LCA is a potent activator of the bile acid receptor, TGR5. TGR5 is expressed in macrophages including KC and has been shown to have anti-inflammatory functions. The authors should, in the discussion, acknowledge the previous work and determine how their data conflicts with or is congruent with previous studies.

Minor:

The authors should provide further information on the FITC-conjugated Nlrp3 probe.

The font size used in several of the bar graphs should be increased.

Author Response

(The authors gave the same response as above.)

Round 2

Reviewer 1 Report

The authors have addressed most of the comments and revised the manuscript accordingly.

As for gender disparity, Suppl Fig 2 is the same as the first submission - No gener information was included. Please provide the data. The request is based on the authors' response that “Response: We checked for gender disparity in BA feeding experiments and did not observed any differences in levels of ALT, AST, ALP and GLDH (Suppl Fig. 2).“

Author Response

We apologize for the inconvenience. In the review process we uploaded the wrong files. Please find attached the supplemental figures.

Round 3

Reviewer 1 Report

The authors have addressed all comments.